# High Amylose-Based Bio Composites: Structures, Functions and Applications

**DOI:** 10.3390/polym14061235

**Published:** 2022-03-18

**Authors:** Marwa Faisal, Tingting Kou, Yuyue Zhong, Andreas Blennow

**Affiliations:** 1Department of Plant and Environmental Sciences, University of Copenhagen, 1871 Frederiksberg C, Denmark; marwa@plen.ku.dk (M.F.); ktings@163.com (T.K.); zhong@plen.ku.dk (Y.Z.); 2College of Physics and Optoelectronic Engineering, Shenzhen University, Shenzhen 518060, China

**Keywords:** high amylose starch, mechanical and physical properties, high amylose applications, polysaccharides

## Abstract

As biodegradable and eco-friendly bio-resources, polysaccharides from a wide range of sources show steadily increasing interest. The increasing fossil-based production of materials are heavily associated with environmental and climate concerns, these biopolymers are addressing such concerns in important areas such as food and biomedical applications. Among polysaccharides, high amylose starch (HAS) has made major progress to marketable products due to its unique properties and enhanced nutritional values in food applications. While high amylose-maize, wheat, barley and potato are commercially available, HAS variants of other crops have been developed recently and is expected to be commercially available in the near future. This review edifies various forms and processing techniques used to produce HAS-based polymers and composites addressing their favorable properties as compared to normal starch. Low toxic and high compatibility natural plasticizers are of great concern in the processing of HAS. Further emphasis, is also given to some essential film properties such as mechanical and barrier properties for HAS-based materials. The functionality of HAS-based functionality can be improved by using different fillers as well as by modulating the inherent structures of HAS. We also identify specific opportunities for HAS-based food and biomedical fabrications aiming to produce cheaper, better, and more eco-friendly materials. We acknowledge that a multidisciplinary approach is required to achieve further improvement of HAS-based products providing entirely new types of sustainable materials.

## 1. Introduction

Polysaccharides are used in a wide array of applications, e.g., food products, biofuels, pharmaceutical industry, and biomaterials. They can be classified at multiple levels depending on their applications such as structure, physical properties, function, and chemical composition. Polysaccharides can be produced from a diversity of sources but are mainly derived from higher plants from which cellulose, starch, and pectin are derived and from animal sources where chitin and chitosan are derived. The diversity of polysaccharide-providing sources that could produce a substance of choice as starting materials that can be modified and used in many applications. The main polysaccharides by volume are cellulose, starch, chitin and pectin (Figure 1) [1].

Starch is one of the most abundant biopolymers on earth is starch. It is widely commonly used in food, pharmaceutical, paper, plastic, textile, and cosmetic industries in liquid dispersions granule, film, or gel forms. Starch is a granular energy reserve composed of amylose (AM) and amylopectin (AP). AM is a mainly linear polymer consisting of α (1,4) linked D-glucopyranosyl units while AP is a branched polymer of α-D-glucopyranosyl units primarily linked by (1,4) bonds with branches resulting from (1,6) linked D-glucopyranosyl units. Most methods for chemical modification of starch depends on the intrinsic reactivity of the hydroxyl groups.

The structure of starch granules can be categorized into four hierarchally levels, ranging of scale from nm to mm (Figure 2). Level 1: Individual branches; this is the distribution of chain lengths and positions of the branches in a sample in the order of 1 nm. Level 2: Whole starch molecules, the structure of the linear and branched molecules are characterized by their molecular weights. Level 3: Nano-lamellar (8–11 nm) structure. The constricted packing of glucan chains facilitating them to intertwine to form helical- semi-crystalline lamellar orders, where AM is considered largely to reside in the amorphous layers. Level 4: Granules; The lamellar structures reside in the inner architecture of native starch granules in concentric growth ring shells of thickness 100–400 nm of different crystallinity creating concentric shells composted of blocklets of different sizes. The final granules are in the range 10–100 μm in size. This level of structure is greatly changed by the process of gelatinization [2].

Based on the botanical origin and genetic background, the composition of AM and AP varies. Starch granules similarly contain various amounts of proteins, lipids, ashes and phosphates. Lipids such as phospholipids tend to form single helical complex with AM, resulting in stability of starch molecule and reducing swelling and water-binding capacity.

The AM: AP ratio, the length of α-glucan chains, and the branching degree of amylopectin define the structure and the functionalities of starch granules from different plant species. The AM present in the starch affects its physiochemical, and functional properties including altered solubility and reduced swelling. Thus, AM can be utilized to modify specific properties of starch making it useful in structure applications.

Starch has been the subject of intensive research over many decades, resulting in a massive number of published studies on physical, biochemical and chemical properties, uses and functionality. In the present review, we aim to provide a concise discussion on HAS applications and forms relevant to its functionality with emphasis on biomaterials and its interactions with other natural polysaccharides.

## 2. High Amylose Starch (HAS)

Chemical and physical treatments are usually used to modify starch functionalities like etherification or esterification to meet the industrial requirements [4]. In addition to physical and chemical modifications, crop breeding and engineering [5], and post-harvest enzyme technology are used as additives in food processing and industrial handling to modify starches [6]. 

HAS is typically directly produced in specific mutants or engineered crops where key enzymes have been knocked out in the crop [7]. For example, an RNA interference line producing high yield of virtually pure AM has been generated [8]. Hence, starch–modifying enzymes can be suppressed or over expressed in the crop to modify starch structure, and functionality. Hence, such genetically engineered crops can be engineered to produce tailor-made starches directly in the starch producing organs by expression enzymes. The potential of using such plants for the production of new environmentally friendly polymers as a part of a sustainable production of plastics is regarded to have an enormous potential [9].

HAS, has a non-optimal structure for generating chain-entangled matrices, which is important for e.g., bioplastics applications, due to the presence of highly branched amylopectin providing too short chain segments for stable double-helical junction zones to be formed [10]. However, the production of pure (99% AM) in a transgenic barley grain system has allowed bulk production for e.g., biomaterials purpose. Direct biosynthesis in the cereal grain can for the first time allow inexpensive bulk production of AM, which until now has not been possible due to too high costs for separation of AM from amylopectin. 

Examples of HAS include the commercially available HAS of corn including Hylon V (50% AM), Hylon Vll (70% AM), Gelose 50 (50% AM), Gelose 70 (70% AM), Gelose 80 (80% AM), and the recently produced barley AM (99% AM). 

HAS gelatinizes at higher temperatures as normal starch due to the long linear AM and AP chains that tend to entangle and crystallize. Due to its unique functionality including high gel forming capacity and resistance to dietary degradation, HAS is an important ingredient for low carb nutrition and raw material for bio plastics production [11]. For future applications in food such as microporous starch, tailored resistant starch, etc., and pharmaceutical materials like controlled delivery systems and biodegradable plastics (starch-based materials), HAS is preferably used.

The linear structure of AM permits complex formation with various host molecules. Generally, the AM inclusion complex acts in the form of a single left-handed helix termed V-type crystalline polymorph as revealed by wide-angle X-ray diffraction that is caused by amylose guest molecular hydrophobic interactions. The guest molecule are included in the cavity of AM helices or in between the helices, depending on the structure of the molecule which suggest the potential of encapsulating of food flavor compounds through interactions with polysaccharides [12].

## 3. HAS-Based Films Production Technology and Protocols

The branched nature of the AP molecules in HAS is the main cause of the well-ordered semi-crystalline nature of the starch granule while amylose is regarded as more or less amorphous. To produce starch films, it is essential to convert the semi crystalline form to a completely amorphous state. Starch is not a thermoplastic material in its native state, but in the presence of a plasticizer e.g., glycerol or sorbitol, water, high temperatures (90–180 °C), and shear it becomes a thermoplastic [13]. Hence, starch loses its semi crystalline granular structure and reach thermoplastic behavior in the production of the thermoplastic starch (TPS). This permits it to be dropped into within various processing protocols including injection molding, casting and blown film processing in a similar manner to traditional plastics.

Starch films can be fabricated using two main approaches based on wet and dry processing. Both methods destroy the granular semi-crystalline form, resulting in quite wide differences in such as permeability and mechanical properties of the films. The dry process depends on the thermoplastic properties of starch induced by extrusion, where starch is heated in low water content 10–40% above its glass transition temperature (Tg). This method is mainly viable for industrial level manufacturing [14]. For the wet process, e.g., for producing starch films, starch is typically gelatinized in excess water, casted, dried. Since, the degradation temperature (Td) is lower than its melting temperature (Tm) [15], plasticizer agents such as polyols and water are used to decrease the intermolecular forces between the starch chains and to form intermolecular bonding between the polymers. Therefore, the mobility of starch chains and thermoplasticity of the starch product improves.

Controlling film texture is a main problem in production of starch films and the use of proper raw material is decisive in this respect. As an example, films made of pure AP (named waxy) become too brittle, due to the resin stiffness resulting in reduced expansions of the blown films. An additional problem is the formation of, double walled films due to sticky extrudates [16]. Alternatively, the native starch can be transformed to a thermoplastic resin, which later can be melt and be extruded using a wide film die in a blown extrusion process [13].

As compared to normal-and waxy starch, HAS requires higher gelatinization temperature and moisture than normal starch to produce films by extrusion. However, as an effect of the high thermo-resistivity, HAS films shows better thermal and mechanical properties compared to their low AM counterparts. This is attributed to entanglement of long linear AM chains and partly retain granular structures in HAS films, which act as a self–reinforcement [13]. Processing features must be adjusted when using HAS-based systems. Hence, different starch films were prepared by extrusion with a focus on the effect on the AM content from the same source. The study conducted on Gelose 50 (50% AM) and Gelose 80 (77% AM). The extrusion processing was problematic due to the high amylose content, which was overcome by increasing temperature and moisture content and regulating the time parameter [13].

To improve the functionality of HAS-based materials, blending with other biopolymers such as chitosan can be performed. For example, a study revealed the extrusion of HAS (70% AM) with chitosan blends generating higher stability with the addition of 5 and 10% of chitosan. The blended extrudates showed a homogenous surface with no agglomeration of chitosan and a reduction in mechanical properties due to chitosan plasticity effect [17].

## 4. Properties of HAS Films

### 4.1. Mechanical Properties

Mechanical properties of bioplastics film including Young’s modulus (E), tensile strength (TS) and percentage of elongation at break (EAB) depend on film composition and nature of components [18]. Starch has poor mechanical properties compared to synthetic polymers. The critical properties of starch are mainly dependent on the high amount of hydroxyl groups providing a highly hydrogen bonding network, but also increasing water exposure [19]. 

The AM:AP ratio is an important factor influencing the mechanical properties of such materials, especially since amylose affects the degree of crystallinity and entanglement. Increased AM content is typically related to an increase in tensile strength and a decrease in strain [19]. This is a direct effect of the linearity of the AM providing entangled chain networks and hydrogen bonded double-helical junctions. However, when preparing HAS-based materials, a significant problem is exactly the resistance to treatment or processing since HAS is more resistant to disintegration by small molecules such as water, which negatively affects its application. 

Brittleness is generally due to the amorphous regions of AM why plasticizers must be included to enhance plasticity. Alternatively, blending with other biopolymers and synthetic polymers such as PVA can improve the mechanical properties while maintaining the biodegradability.

The tensile strength increases as AM content increases although this relation is not linear [20]. Factors such as plasticizer type and content, crystalline structure, granular remains in the matrix and the extent of plasticization of the amorphous parts all affect the mechanical properties of a bioplastic model. Moreover, the protocol used for preparation of the films affects the mechanical properties. For example, casting and preparation using microwave heating or adding alkali shows better mechanical properties as compared to the treatment under high pressure and temperature. The difference in the tensile strength of the three methods are associated to the leach of AM and formation of AM-lipid complexes with the semi-crystalline and lamellar structures during heating and retrogradation [21]. More AM leach out form the granules in the microwave heating method, since microwave energy efficiently promotes rupture of the starch granules, while the integrity of granular starch is higher under pressure heating and alkali treatment [22,23]. 

The presence of nano-fillers e.g., nano-clay in Hylon VII enhances the mechanical properties, which is credited to the mechanical resistance exerted by the nano-clay itself. Theoretically, the complete dispersion of the clay as nano-layers in the polymer optimizes the number of available reinforcing elements for carrying an applied load. The link between the large surface area of the clay and the polymer matrix facilitates stress transfer from the matrix to the nano-filler, resulting in this tensile improvement [24].

### 4.2. Plasticizing Agent Effect

Plasticizers are used to increase the flexibility of the films as an effect of increase in the molecular mobility and decrease in the degree of crystallinity and Tg. Plasticizers such as polyols are commonly used due to their low molecular weight and hydrophilic properties. Polyols have an affinity to water molecules depending on the number of the hydroxyl groups present and their molecular weight. However, the mechanical properties of systems including polyols are difficult to predict [13]. The range of glycerol and sorbitol content mostly used is in the range 20–50% *w*/*w* and insufficient content of these plasticizers may lead to an anti-plasticization effect, e.g., adding 15% of glycerol to barley AM-only (99% AM) composite films reinforced with cellulose nanofibers (CNF), shown reduction at EAB. This phenomena was reverted by adding 25% of glycerol which significantly increased the EAB [10]. Extruded AM derived from the barley AM films showed higher mechanical properties as compared to normal starch, both at 30% glycerol content [25]. The TS of the AM was 6-fold higher and 2-fold higher at EAB compared to a control normal barley starch. 

Addition of more than one plasticizer other than water in the system can result in stronger plasticizer interaction, which reduces film dryness and enhances particular properties. Glycerol (3-hydroxyl groups) and xylitol (5-hydroxyl groups), tested separately and in combination with ratio 1:1 on high AM (80%) and normal AM (25%) films [26], showed that xylitol was more effective due to its larger molecular size and strong tendency to form stronger hydrogen bonding with the glucan hydroxyl groups. Xylitol-plasticized films exhibited higher moisture migration fluxes and effective diffusivity values. The HAS-glycerol films kept the highest glass transition temperature, moisture content, higher TS, higher modulus of elasticity and lower EAB among all films. Regardless of the starch type, plasticizer concentration above 15% showed lower mechanical properties. Besides, all films showed brittleness upon drying due to the antiplasticization behavior (Table 1). Films with glycerol had higher moisture content compared to xylitol. When combined, the moisture content was decreased due to the stronger starch-xylitol interactions allowing increased water evaporation. HAS films with xylitol also showed high crystallinity due to the high moisture indicating that xylitol crystallizes at sufficiently high moisture content. The starch films by combined plasticizers did not exhibit anti-plasticization providing an argument for combining plasticizers [26]. 

At lower AM content and with 30% glycerol, the TS of HAS (30% AM) films was higher than the TS of HAS films containing (50% AM). However, using 30% of urea/formamide as plasticizer, reversed the mechanical results, e.g., the TS of 50% AM films was 2-fold higher than films containing 30% of AM [27].

A comparative study investigated the effect of the ionic liquid 1-ethyl-3-methylimidazolium acetate ([Emim] [OAc]) as a plasticizing agent and glycerol on using Gelose 80 (AM 82%). Gelose 80 is known to be very resistant towards gelatinization and thereby shows low processability. The study showed that [Emim] [OAc] was effective in a compression moulding process involving only minor shear treatment. In addition, scanning electron microscopy (SEM) analysis showed that [Emim] [OAc], as compared to glycerol, contributed to less granular remains. Hence, [Emim] [OAc] at low concentration efficiently disrupted the original granular B-type and V-type crystalline polymorphs and also prevented single or double helix formation is due to the strong interaction between starch hydroxyl group and acetate anion in [Emim][OAc] thus producing less V-type crystalline structures and increased the mobility of the amorphous starch [28].

### 4.3. Barrier Properties

Starch films have excellent oxygen barrier properties due to their high ordered hydrophilic hydrogen-bonded network structure, in which AM and AP form partly tightly connected crystalline networks. Differences in the molecular arrangements of the starch, plasticizers, the microstructure, thickness of the films and different storage conditions has tremendous effects on the barrier properties. Generally, plasticized starch-based films show poor water vapor permeability (WVP) due to their hydrophilic nature and increases water diffusion rate within the film (Table 2).

In order to understand the permeability function of HAS-based films, other parameters must be provided such as thickness, plasticizer content, and temperature and water activity [29]. As expected, increasing film thickness causes increased resistance to mass transfer across. Hence, the equilibrium water vapor partial pressure at the internal matrix increased with film thickness [30]. The water vapor permeability (WVP) of HAS (70% AM) films without plasticizer was 11.7 × 10^9^ g/ms Pa and it increased to 14.7 × 10^9^ g/ms Pa with increased concentration of plasticizer. Films with glycerol had higher WVP than those of sorbitol [31]. However, adding the hydrophobic maize protein zein to HAS decreased the WVP to 8.4 × 10^9^ g/ms Pa due to the inherit hydrophobicity of maize zein itself [31].

These results are substantiated using plasticized Amylomaize (65%AM) used as a coating material for strawberries to extend their shelf life [32]. The coating system was formulated with, without plasticizers, and with lipids such as sunflower oil to improve the barrier properties and decrease weight loss of the strawberries. Without plasticizers, the Amylomaize had lower 2.62 × 10^10^ g m^−1^ s^−1^ Pa^−1^ WVP than normal maize starch 3.68 × 10^10^ g m^−1^ s^−1^ Pa^−1^. The plasticizers used were glycerol and sorbitol the latter showing lower WVP values (Table 2). The presence of sunflower oil reduced the WVP and did not modifyO_2_, CO_2_ permeability compared to the plasticized starch films. The effect of plasticizer type was diverse for the O_2_ permeability, i.e., using sorbitol reduced the permeability than glycerol. 

An additional way to control the permeability and mechanical assets of a starch film system is to employ a nanocomposite approach (Table 2). For a pure AM/CNF film, system the AM/CNF composite films without glycerol showed a reduction in the WVP. Increasing CNF concentration the WVP increased due to the influence of dense network structure formed by CNF, that generates a more tortuous diffusion path for gases [33]. Hence, the pure AM films showed a 7-fold higher WVP than CNF films, the WVP of all the tested composite films were far lower than the majority of petroleum-based materials tested [4]. Addition of montmorillonite (MMT) nano-clay with Hylon V and HylonVII increases the tortuosity of the diffusive path for a penetrating molecule. The dispersion of nanoclays in the starch matrix depended on the compatibility and the polar interactions among the starch, the silicate layers, and glycerol. The starch/MMT composite films showed higher TS and better barrier properties to water vapor due to the formation of intercalated or exfoliated nanostructure, however, the WVP did not change significantly when AM > 50 [24].

Moreover, using different types of plasticizers in combinations affect the WVP. Combining xylitol and glycerol at 20% combined concentrations using a HAS (80% AM) film system resulted in lower WVP than using only one of the plasticizers. However, doubling the plasticizer concentration increased the WVP values [26].

**Table 2 polymers-14-01235-t002:** Barrier properties of plasticized HAS films.

Starch	AM%	Plasticizer %	WVP × 10^10^ (g/ms Pa)	Gas Permeability O_2_ × 10^10^ (cm^3^/ms Pa)	CO_2_ × 10^9^ (cm^3^/ms Pa)	References & Remarks
Amylomaize	65	0	2.62	28.05	26.45	[32] Gly and Sorbitol concentrations 20%
		Gly	2.14	3.21	3.85	
		Gly + SO	1.76	4.39	2.36	
		Sorbitol	1.21	2.96	2.28	
		Sorbitol + SO	0.97	3.43	2.18	
HAS	80	0	0.52	ND	ND	[26] Xylitol and glycerol concentrations 20%.
		G	0.43	ND	ND	
		Xylitol	0.11 (g mm/m^2^ h kPa)	ND	ND	
		Gly + Xylitol	0.14	ND	ND	
AM-only	99	0	0.351 (cm^3^ mm Pa/m^2^ 24 h)	ND	ND	[10]

Gly: Glycerol; SO: Sunflower oil; ND: not determined.

## 5. Forms of HAS

### 5.1. Foams

Many attempts have been made to commercialize and recycle materials from renewable resources such as starch to substitute expanded polystyrene (Figure 3). Starch-based foams are brittle and sensitive to water resulting in expensive coating steps when it contacts to hot or cold aqueous liquids. Moreover, the AM: AP ratio is important for the foaming properties in place of the accessibility of water during the process (Table 3). Hence, chemical modification and additives are investigated in baked foam plate preparations to enhance water resistance and strength. Extruded native starches with AM content of 50% display some preferred properties [34]. However, starches with high AM content expand less due to the entanglements between linear AM chains [35].

Baked starch foam trays using maize starches with different AM content ranging 18%, 27%, 50 and 70% AM, revealed that the foam strength and density increased with both the starch concentration and AM content. However, foam flexibility tends to increase at lower density. Foam plates made from potato starch exhibited higher flexibility and lower density than those prepared from maize. Swelling and water holding capacity are the key to bubble formation since it gives the paste an elastic behavior and creates a gel which efficiently traps the steam bubbles and create pores during the process. Using existing protocols, HAS is not completely gelatinized when bubbles are introduced to the gel and does not have strength to trap the bubbles and create the pores [34]. Foams produced from different potato starches in granular form with varying amounts of AM content using a microwave approach demonstrated that the HAS prototypes produced compact structures. The dry matter content during the microwave treatment showed that the water evaporated more rapidly in the HAS foams than in the normal and pure AP from potato starches [36]. 

Trays prepared from chemically modified starches are less dense, requires shorter baking times and shows higher EAB values than trays prepared from unmodified starches. Plates made of polyvinyl alcohol (PVOH) and HAS (50% AM), had higher EAB at low humidities than those made from normal starch and PVOH. Addition of softwood fibers increased the strength of the starch foam plates at both high and low humidities. Addition of monostearyl citrate to starch formulations provided the best improvement in water resistant among the compound tested [36]. Foam plates made from HAS (50% AM) were shaped irregularly and very dense as an effect of incomplete starch gelatinization and the absence of paste elasticity. Plates made from acetylated HAS with (50% AM) had regular shapes, smooth surface, less dense and baked in much shorter times than plates made from unmodified HAS starch. Plates were not well formed from hydroxyl propyl Hylon VII (70% AM) as viscosity and elasticity of the paste were too low to expand foam using the baking process [36].

Extruded foams were prepared from maize starch (70% AM) with and without sodium stearate and PVOH to form an AM-sodium stearate complex at low temperature. In the lack of PVOH, water absorption and foam shrinkage were decreased at 95% RH due to the hydrophobicity of the amylose-sodium stearate complex. Addition of PVOH increased the expansion ratio although the shrinkage at 95% RH was still less than that observed with the uncomplexed AM. Besides enhancing the tensile properties this procedure permits manufacturing of biodegradable starch foams for specific end-use applications [37].

To enhance the performance and the hydrophobicity of the foam, adding of corncob fiber particles (1 mm maximum particle size) and cellulose to HAS-acetate with (70% AM) extrusion was conducted in a twin-screw extruder at 160 °C barrel temperature and at 225-rpm screw speed. Significant differential functional properties were revealed for the different HAS-acetate/corncob foams. The bulk densities increased with increasing corncob content and the highest compression strength was obtained with 20% corncob and 13% ethanol content as a solubilizing agent. Blending with corncobs slowed radial expansion of starch acetate foams, whereas cellulose did not affect expansion. Water absorption isotherms showed good hydrophobic properties after adding corncob and cellulose foams [38].

### 5.2. Starch Fibers

Starch fiber mats are produced by wet electrospinning, which is a technique to produce micro to nanoscale fiber mats with high porosity, high surface area, and several other functional nanostructures. Hence, to explore such systems, starch fiber mats with random and different orientations have been established (Figure 3). The unique properties and features produced by electrospinning enable starch and other biopolymers to be applied for a wide array of different biomedical application such as tissue engineering, wound dressing, and drug delivery (Table 3). HAS are favored for electrospinning over low AM starch types due to its molecular linearity, and AM can be readily electrospun into fibers (Table 3). Pure HAS starches have been effectively electrospun into fibers using solvents such as formic acid, dimethyl sulfoxide (DMSO) or a mixture of (DMSO/water). 

Aqueous formic acid (60–100%) was used to dissolve Hylon Vll (17%) to yield a spin dope solution for electrospinning of HAS into pure HAS fibers of 80–300 nm in diameter. Formic acid played an essential role by disrupting the starch granule structure by esterification by generating mono-formate esters at the C6 position of the glucose units in the starch [39]. The resulting HAS-based fibers exhibited higher EAB (26%) at a high concentration of formic acid without preferable molecular orientation compared to normal starch highlighting the potential of electrospun HAS fibers as a low-cost and sustainable biomaterial appropriate for food and pharmaceutical applications. 

In another test, Hylon VII-acetate fibers were formed with different degrees of substitution (DS) generated with acetic anhydride followed by dissolving the modified polymer in either formic acid/ethanol or formic acid/water solvent. The resulting fibers were 50 µm with highest tenacity of 17.9 MPa [40].

The stability of electrospun fibers in aqueous media can be enhanced by post spinning treatment in ethanol 70% for 1 h to increase starch crystallinity or crosslinking with aqueous glutaraldehyde using DMSO as a solvent. The produced cross-linked fibers had excellent water stability as compared to the native starch as an effect of the crosslinking. The TS of the nanofibers web was 10 times higher than native starch [41].

The alignment of two-dimensional structures like fiber mats is determined by a multitude of raw material as well as processing factors. Aligned HAS fiber mats were successfully produced using Gelose 80 by wet electrospinning with controlled parameters like drum location, rotational speed, and coagulate bath composition. The selected parameters influenced the alignment fiber formation. The TS was affected by the concentration, location, and rotation speed of ethanol. Improved coherent fiber alignment was produced at higher rotational speed and lower ethanol concentration (60 and 80%) but not at 100% ethanol. TS was correlated with better fiber alignment, the weight–normalized ultimate TS (WNUTS) values ranged from 13.55 to 87.74 N/g among all treatments [42]. 

Preventing or slowing the re-association of AM is crucial for electrospinning HAS from an aqueous dispersion. Producing AM-guest inclusions is one option to delay such re-association. Blending 8.5% of Gelose 80 with 4% pullulan and 6.5% sodium palmitate (% of AM) optimized the solution for electrospinning [43]. A high-temperature process was applied to disarrange the recalcitrant Gelose 80 structure. It was demonstrated that pullulan prevented starch association (retrogradation) and the sodium palmitate improved the HAS stability in water by slowing the recrystallization process. Nanoscale-sized fibers 146 ± 50 nm were obtained with TS levels resembling ultra-light weight fiber materials (Table 3). Producing electrospun biopolymer fibers from aqueous dispersions showed their potential use in cosmetic, food, and pharmaceutical areas.

### 5.3. Gels

Hydrogels are polymeric materials with three-dimensional network structures that are capable of swelling in aqueous media. These are widely used in the control release of compounds due to the facility of dispersion in the hydrogel matrix. The properties of a given hydrogel depends on both on external conditions and the physiochemical properties of macromolecules [44]. Starch gel properties are especially influenced by the AM:AP ratio. The HAS gel formation was faster at 150–152 °C, while gelatinization decreased at higher temperature 140–165 °C. Heating at higher temperatures caused extensive degradation of AP which affects on the final gel formation properties of HAS (Hylon VII) and leads to phase separation that was visible by light microscopy [45]. For gels produced from the gelatinized starch, high content of AM provided hard and deformable gels with low water holding capacity. In vitro, amylolytic digestion assays showed that gels with high AP released more glucose and were more efficiently degraded than high AM gels. The non-gelatinized HAS starches gel was suggested to be suitable for encapsulating compounds for delivery in the colon or for use as dietary fibers [46].

In order to produce long shelf life of pasta/noodles products through retort processing, gels were produced by mixing starch and protein to form composite gels. To permit protein crosslinking, microbial transglutaminase (MTGase) was mixed along with Hylon VII and wheat flour at different levels and at high temperatures. It was shown that increased Hylon VII content increased the firmness while the adhesiveness decreased. The combination exhibited harder gels where cross-linking network formed by MTGase enhanced the gel elasticity and structure compared to wheat starch alone at 121 °C (Table 3) [47].

The effect of hydrocolloids such as xanthan gum and guar gum on gel properties of HAS e.g., Hylon V (50% AM) and HAS (71% AM) showed that the presence of gums in HAS resulted in increased viscosities and suppressed swelling of starch granules as compared to a pure AP starch [48]. Guar gum had a stronger influence on the gel strength and pasting properties on HAS as compared to xanthan gum, which may be attributed to the stronger molecular association between guar gum and AM. Further studies are required to study the interaction between hydrocolloids and HAS for further applications [48].

## 6. Uses of HAS

### 6.1. HAS Complexes

Relevant for direct food applications is the capability of AM in HAS to form inclusion complexes with variety of flavor compounds. Importantly, the starch provides a matrix showing a decrease in flavor perception due to the starch-flavor inclusion complex formation. Size, hydrophobicity, water solubility and shape of the ligand contribute to the ability of the ligand to complex with starch [49]. Interactions at high concentration of aroma compounds such as citral, butyric acid or octanol with native starch granules can induce major disruption of the granular structure [49]. A ligand with a long chain hydrocarbon induces complexes of single helical AM with the hydrocarbon included in the hydrophobic channel (Figure 4) [50]. It has been reported [51] that shorter linear molecules such as C-8 fatty acid or C-7 lactone cannot form stable complexes with starch. Lipids present in the native HAS granules may enhance complexations with low water solubility compounds by forming ternary co-inclusion complexes of starch lipid flavor (Table 3) [51].

Wide-angle powder X-ray diffractograms of Hylon VII-fatty acids (palmitic, oleic, steric, myristic, and capric) complexes prepared at 30 °C, 50 °C, 70 °C displayed the typical V-type crystalline polymorph, the size of the formed crystals decreased, and the crystallinity degree increased as the preparation temperature of the complexed AM increased. SEM micrographs showed that presence of complexed crystals either in the form of lamella or in the form of spherulites. For example, thermogravimetric analysis (TGA) data (solid state) indicated that oleic acid was adequately protected in the form of complexes [52].

Proteins has also an effect on AM-lipid complexes. Such multiple interactions are likely to be common in cooked starch-based foods and may influence their functionality but studies are limited [53].

As opposed to fatty acids and alike, HAS nanoparticles showed lower encapsulation efficiencies (45%) towards anthocyanins compared to AP and normal starch nanoparticles (52 and 49%, respectively). AP molecules with low degree of substitution favored the encapsulation efficiencies with anthocyanins by succinaylated nanoparticles. The nanoparticle-anthocyanin interaction occurred through electrostatic and hydrophobic staking interactions. The nano-capsule obtained from AP had the smallest particle size (<100 nm) [54].

### 6.2. HAS Based Food Products

Bakery foods are starch-rich, which provide a good texture for consumers but are typically highly glycemic by containing high amounts of rapidly digestible starch (RDS) and slowly digestible starch (SDS), and a low amount of RS. Especially, high amounts of RDS has a great potential to induce an excessive postprandial glycemic response, obesity and cardiovascular disease. HAS flour can act as an efficient and highly palatable ingredient to substitute wheat flour (WF) for the preparation of novel starch-based food to improve RS content, and decrease glycemic index. For example, HAS (50% AM) added to cookies showed 10.1% of RS compared to 1.6% in a pure WF cookie [55]. Bread consisting of HAS (50% AM) flour improved the RS content from 1.5% to 3.1% [56]. To explore HAS effect on texture, morphology and volume on food, different ratios (0–40%) of HAS (56% AM) were tested in a cake. Sensory assessment results showed that the consumers accepted 20% of HAS in cakes and without negative effect on physiochemical properties of the cake. Hence, HAS has a potential as an ingredient in food to develop the quality of nutrition and texture (Table 3) [57].

AM-only (99% AM) produced in barley exhibits low predicted glycemic index (pGI) due to suppressed starch digestibility. Substituting wheat flour with AM-only flour in a bread model provided a highly microbiota-modifying bread model with reasonably good baking properties (Table 3) [58].

### 6.3. Applications of Starch-Based Films in Food Applications

Currently, many applications such as food packaging and drug delivery increasingly replace non-biodegradable materials with biodegradable natural raw materials. HAS is especially relevant as a safe, odor-less, robust, and biodegradable raw material for food packaging and food coating (Figure 3). Especially, the high oxygen barrier properties of starch-based films offer a very good option for edible coating for foods with high respiration rates i.e., vegetables and fruits. Such films combined reduce respiration and delay oxidation. Due to the high hydrophilic nature of starch and its resulting high WVP, starch is a preferred choice, especially when the coating is required to be washed off. HAS (65% AM) coating provided a higher effect in generating a rigid network leading to a smaller extent of plasticization and increasing the stiffness of the films as compared to normal starch (25% AM) coating (Table 3) [59].

To improve mechanical and barrier properties, several materials have been incorporated into HAS-based films resulting in blended composite films. The incorporation of cellulose nanocrystals (CNC) has been shown to provide a more homogeneous surface and increases the amorphous halo and reduce the crystallinity, which confirms the interaction of nanoparticles to glycerol molecules and limits AM-glycerol interaction. Dynamic mechanical analysis (DMA) of the HAS/CNC films suggested a hydrogen-bonded cellulose network within the starch matrix. Furthermore, the incorporation of CNC can shift the start of material degradation to higher temperatures [60]. Casting composite films of HAS (80% AM) with CNC increased the Young’s modulus, and the TS along with reduction in the EAB when CNC reached 10%. The transparency rate was reduced after addition of cellulose crystals due to the formation of a two phase system, which can improve the property of protection against UV radiation which is important for drug and food packaging [61]. Another starch-cellulose system tested was the blending of AM with cellulose nanofibers (CNF). Such a blend showed increased hydrophobicity and superior mechanical and barrier properties as compared to the pure AM system especially at 25% of CNF due to the strong interaction between α- and β-bonded polysaccharides (Table 3) [10].

Another example is the effect of methylcellulose (MC) 6% (*w*/*w*) as an enhancer to film preparation of chitosan-HAS (55% AM) in addition to glycerol. MC increases the film permeability and mechanical properties. Excessive increase of MC declines both the mechanical and the barrier properties due to the heterogeneous structure composite (Table 3) [62].

Plasticized composite films based on HAS (85% AM) and konjac glucomannan (KGM) were developed to enhance morphological irregularities, which is typical for pure HAS-based packaging materials. The similarity in the linear structure of KGM and HAS makes it rather compatible with HAS. The HAS/KGM composite films were fabricated at KGM concentrations of 0.1–0.5% (*w*/*v*). At lower concentrations, phase separation was the main effect of KGM on HAS films while the linkage inhibition developed at high addition concentrations of KGM. The WVP values increased with more of KGM. This could be due to the incompatibility of the two polymers. The WVP result suggested that the composite film maybe not be suitable for application as high barrier film. The optimal content of KGM was 0.3%, where a homogeneous and compact structure was formed (Table 3) [63]. Interestingly, adding β-cyclodextrin (β-CD) increased the number of double-helical junction zones between starch segments particularly AM chains enhancing hydrogen bonding thereby enhance the strength of the molecular network [64]. The incorporation of β-CD into the composite film helped to reduce the phase separation, promote the interaction between the polysaccharide chains and enhancing the mechanical and barrier properties. The TS values reached 13.77 MPa after adding 1.5% of β-CD as compared to 10.27 MPa for the HAS-only system (Table 3). 

For HAS (80% AM)/Chitosan plasticized films, TS values (3.44 MPa) were lower than that for HAS/KGM composite films [65]. The EAB values increased with KGM which was suggested to be attributed to weak interaction between KGM and HAS, a result that was similar to the addition of chitosan to HAS maize starch [66]. Chitosan attracts strong research interest because of its biodegradability, non-toxicity, and biocompatible cationic assets. Blending starch with chitosan has been demonstrated to improve mechanical properties, reducing WVP, and enhance antibacterial properties. To improve the miscibility of chitosan and HAS (80% AM) by mechanical shearing using micro-fluidization was applied. As compared to the elastic HAS/chitosan composite films, the pure HAS films were rigid and brittle without glycerol. Including glycerol, these blends enhanced the hydrogen bonding interactions between the two polysaccharide constituents. An anti-plasticization effect was observed in composite films when the glycerol concentration was 2.5% (*w*/*w*) which was reflected in visible surface cracking in the SEM image. At a glycerol concentration of 5%, the thermal properties, and the TS were reduced and the EAB increased with increased glycerol content (Table 3) [65].

HAS (72% AM) was used as a filler to enhance the gelation of gelatin films under different gelatinization conditions and at different concentrations. The presence of HAS increased the film integrity, transparency, thermal properties, water solubility, and thickness. Mechanical properties were improved with the gelatinized starch especially at 50% concentration. Water resistance was observed when HAS was gelatinized in hot alkaline media. The outcomes may serve as a platform to improve gelatin film performance and potentially extend applications of edible gelatin films in the food industry (Table 3) [66].

### 6.4. Antimicrobial Food Packaging

As described above, HAS films are biologically absorbable, non-toxic, and are good barriers to oxygen and semipermeable to carbon dioxide. Incorporation of active substances such as antimicrobial agents, plasticizers and lipids into HAS-based matrices allow controlling of the release behavior of the amount and the diffusion rate of the active substances [51]. Incorporation of 1% grape seed extract (GSE) into pea starch (naturally high in AM) films influenced bacteriostatic effect against meat microbial surface load during the first 4 days of incubation. The migration rate of the phenolic compounds into the meat surface play an important role in the inhibition of Bacillus thermosphacta. The slow-release rate is due to the interaction between the hydroxyl groups of phenolic compounds and starch. Besides the migration of the phenolic compounds in GSE might be physically entrapped among the AM helical matrix. GSE films presented remarkable advantages for marketability as they inhibit the growth of undesirable pathogens in meat, improving meat quality and extending its shelf life [67]. 

Hydroxyl propyl-HAS (80% AM) films functionalized with different contents of pomegranate peel (PGP) powder can be potentially used as a reinforcement and as an antimicrobial agent. All parts of pomegranate are widely reported as antioxidant as well as antibacterial. The pomegranate fruit peel being rich in phenolic compounds such as catechin, rutin and epicatechin [68]. While the peel is mainly contains of pectin, cellulose, lignin, and some proteins. Films of HAS-PGP blends was demonstrated to enhance mechanical properties as compared to pure HAS films and also showed good antimicrobial effect against both S. aureus and Salmonella spp. Increasing the PGP content, increased the inhibition zone of the films against targeted microorganisms [69].

### 6.5. Application of HAS as a Biomedical Material

As previously mentioned, most of the native starches are limited in their direct application due to it suboptimal mechanical properties, high water sensitivity, and poor long-term stability. This is also valid for HAS. To overcome these limitations HAS blended with other raw biomaterials may lead to new biodegradable biomedical scaffolds with potentially better properties that can be used in advanced applications such as within the medical and pharmaceutical fields (Figure 3).

A comparative study investigated the properties of collagen with three different maize starches: waxy starch (0% AM), normal starch (27% AM) and HAS (72% AM). Blending collagen with starch at both low (10%) and high (50%) concentrations enhanced the mechanical strengths, HAS provided the highest strengths at both concentrations. AM small uniform granule created less heterogeneous films, providing strong interaction with the collagen fibers, whereas larger granules such as those found in the waxy and normal starch preparations disrupted the continuity of the collagen matrix and decreased the cohesive force between collagen fibers, which eventually decreased the mechanical strength. Films of waxy and normal starches were degraded following heating while HAS (72% AM) still maintained its granular shape. The addition of HAS increased the thermal stability, and reduce film solubility in water. Hence, HAS improved collagen film, providing a potential solution to address challenges facing e.g., collagen-based casing industry (Table 3) [70].

On the contrary, physical properties of extruded starch-based materials with different glycerol concentrations were examined using starch from two different botanical origins, amylomaize (70% AM) and potato (20% AM), after immersion in a physiological medium, as well in vitro cell effect and in vivo tissue response in a rat animal model. Potato starch samples with glycerol 20% and HAS (70% AM) samples without glycerol showed an elastic modulus of 1.5 and 9.8 MPa and only weak swelling in water, lasting stable for 30 days. Stabilization of the mechanical properties and geometry was due to the evolution of crystallinity in potato starch samples. The crystallinity of the potato starch samples with 20% glycerol increased the B-type crystallinity from 15% to 25% after 30 days of immersion. While the B-type crystallinity of HAS samples were disrupted by amylose-lipid complexes that led to the superimposition of the V-type pattern [25].

In vivo implantation materials, the potato starch-based prototypes were integrated into the tissue with a limited inflammatory reaction, due to the presence of M2 macrophages that is generally allied with the implant healing. However, the HAS material induced a pathological foreign–body reaction at day 30 post implantation as revealed by granuloma formation, the presence of giant cells and macrophages going towards HAS degradation [71]. This study revealed crystallinity as a key parameter of new degradable materials. Hence, plasticized potato starch could be a new candidate for biomedical applications due to its good tissue integration and durability.

For bio-therapeutic applications, electrospun starch-based, core-sheath composite fibers, have been demonstrated to be promising delivery vehicles with the aim to improve the stability of live bacteria. The use of such composite fibers for encapsulation of viable probiotic cells showed that these fibers with mean diameters of 4.13 μm could be used as carriers to stabilize Lactobacillus paracasei cells, where the glycerol core can act as the cell suspension medium. The sheath solution was composed of 17% Hylon Vll dissolved in formic acid to produce starch formate. The core-sheath starch-formate/glycerol composite fibers were capable of retaining bacterial viability when stored at 4 °C and room temperature for up to 21 days [72]. 

Considerable efforts are made to develop colon targeting drug systems, aiming at securing direct treatment of colon-related disorders and providing more precise strategies for chronotherapy for diseases that are sensitive to e.g., the circadian rhythm [73]. Especially, the delivery of peptide, and protein-based drugs to the systemic circulation through colonic absorption has gained recent interest due to the relatively low proteolytic activity and longer retention time in the colon. For this purpose, AM in HAS is of special interest, because its resistance to pancreatic α-amylases can ensure sufficient time for a solid oral dosage to reach the colon [73,74,75]. The early coating delivery systems with AM required extraction of AM from starch e.g., as a butanol complex prior to coating (Table 3) [76]. 

Improvements included heating of HAS to 80 °C during the aqueous film coating process (WO 2008/012573 A1), which prevented the release of drugs in the stomach or upper intestine, while the release was observed in the colon as an effect of specific degradation of the starch coating by the intestinal microbiota [77]. Hydroxypropylated, acetylated, and pregelatinized HASs have also been studied as coating materials for colon targeting vehicles. These starch types in combination with ethyl cellulose were found suitable for the preparation of a variety of coating systems to enclose peptide- and protein-based drugs (Table 3) [78]. Hence, HAS is a viable raw material for colon-specific delivery systems. However, these systems are not yet application-ready and further improvement of the properties of HAS film coatings and clinical trials are required.

Recent efforts in the bone and tissue engineering field have been made to form bone scaffolds that mimic the structure and function of the bone. Preliminary examination of the effect of the chemical constitution of the composite was performed using hydroxyl apatite and printed by a solid freeform fabricator (SFF). Compared to cassava (15% AM) and potato (20% AM) starches composites with hydroxyapatite, corn starch (25% AM) improved the mechanical strength from 2.12 ± 0.77 MPa to 12.49 ± 0.22 MPa, which is more closely to the mechanical strength of natural cancellous bone [79].

High AM content in the starch facilitates the hydroxyapatite to interact with the starch particles with hydrogen bonding. AM was proven to provide a biologically active reinforcement moiety of resorbable bone scaffolds through an interparticle and crystal interlocking mechanism (Figure 5), improving the mechanical integrity of the composite scaffolds for bone replacement applications (Table 3) [79].

**Table 3 polymers-14-01235-t003:** Forms and applications of HAS.

Starch	AM%	Method of Preparation	In Combination with	Forms and Application	Remarks	References
Amaizo5 Amylomaize	50 70	Baking mould		Foam	AAM content increased: density, foam flexibility decreased, trapped air bubbles, more pores produced.	[35]
HAS	50	Baking mould	PVOH Soft wood fibers Monostearyl citrate	Foam	Strength increased at high and low humidities. Heavy, with irregular shapes.	[36]
HA-acetate (Acetate high amylose starch)	50	Baking mould	Guar gum Magnesium stearate	Foam	Lightweight with regular shape.	[36]
HP-HylonVII (Hydroxyl-propyl Hylon VII)	70	Baking mould		Foam	Viscosity and elasticity of the paste were too low to expand foam using the baking.	[36]
HAS	70	Extrusion	PVOH Sodium stearate	Foam	Reduce shrinkage at 95% RH. Enhanced tensile properties.	[37]
HA-acetate	70%	Twin-screw extruder	Corncob Cellulose	Foam	Corncob and cellulose enhanced hydrophobic properties. Bulk densities and strength increased.	[38]
HylonVII	70%	Electro spinning	Aqueous glutaraldehyde 50% (GTA)	Fibers Diameter (200–700 nm)	Stable in water, non-toxic, 10 times more strength than uncross linked fibers.	[41]
Hylon VII	70%	Electro spinning	Acetic anhydride	Fibers Diameter (50 ± 5µm)	Uniform fibers obtained with small diameter at formic acid concentration (90%).	[40]
Gelose 80	~80	Electrospinning		Fibers diameter between (2.15–4.02 µm)	Better alignment occurred at higher rotational speed and lower ethanol concentration. Speed.	[42]
Gelose 80	76	Electrospinning	Pullulan, Sodium palmitate	Fibers, Diameter (146 ± 50 nm)	Pullulan hindered starch association. Tensile strength of the nanofiber composite was found to be weaker than that of micro-sized pure starch fiber mats.	[43]
HylonVII	70	Electrospinning	Dissolved in different formic acid dispersions (FA)	Fibers, diameter (304 nm) at 100% formic acid and 84 nm at 80% FA.	Diameter decreased as water content increased	[39]
Hylon VII	70	Special pressure vessel at 140–165 °C		Hydrogel	Lost its rigidity, due to the degradation of AP	[45]
Hylon VII	61	Mixing gelatinized starch then autoclave treatment	Alginate matrix	Macro gels	A high AM amount in the starch the produce gels with less degradation after digestibility compared with common starches and high AP starches.	[46]
Hylon VII	−70		MTGase	Gel	MTGase treated gels can withstand high temperature. Hylon VII added to the gels supplied tighter, stronger, and denser protein network.	[47]
Hylon VII	−71	Heated in high pressure reactor apparatus	Guar gum/Xanthan gum	Gel	Guar and xanthan gums affected the pasting properties of normal maize starch more than those of waxy maize starch. no new covalent bonds were formed between the guar and xanthan gums and the starches (normal, waxy and high-AM).	[48]
Amylomaize VII	70	Starch cold gelatinized	Glycerol	Coating	HAS, coating reduced strawberries weight loss and decay. Maintain freshness compared to medium AM starch.	[59]
HP-HAS (hydroxyl propyl high AM starch)	80	Blending and casting	Cellulose crystals Glycerol	Composite film	Improved transparency and mechanical properties.	[61]
HAS	55	Blending and casting	Chitosan, Glycerol, Methyl cellulose	Composite film	Permeability of gas and water increased. Mechanical properties decreased.	[62]
HAS	85.5	Blending and casting	KGM, glycerol	Composite film/Packaging film	Phase separation, high WVP permeability	[63]
HAS	85.5	Blending and casting	KGM, glycerol, β-cyclodextrin	Active composite packaging film	Enhanced WVP, Mechanical properties. Reduced moisture content.	[64]
HAS	80	Blending and casting	Chitosan, glycerol	Composite film	Anti-plasticization effect at 2.5% glycerol accompanied with visual cracks.	[65]
AM-only	99	Blending and casting	CNF, glycerol	Composite film/ Food packaging	Better mechanical properties. Anti-plasticization at 15% of glycerol.	[10]
Hylon VII	−70		Thymol, Menthone Limonene cymene	Starch –flavor complex preparation.	HAS for flavor encapsulation by inclusion technique, effectively entrapped low water solubility flavors.	[51]
Amylomaize	−56	Formation of V-AM molecular inclusion complexes	Capric Myristic Palmitic Stearic oleic	Starch –flavor Complex	Oleic acid in the form of Hylon VII starch complex is efficiently protected against oxidation as well as thermal degradation for at least up to 100 °C	[52]
HAS	72	Blending and casting	Bovine-hide gela- tin, type A. 0.1N NaOH	Composite film	Thickness and transparency increased. Enhanced mechanical properties and water solubility of gelatin films	[66]
HP-HAS	80	Blending and casting	Pomegranate peel (PGP), 20% ethylene glycol	Anti-bacterial, edible composite film/Food industries	It was found that the developed films demonstrated good antibacterial properties against both *S. aureus* and Salmonella, and enhanced the mechanical behavior.	[69]
HAS	72	Blending and casting	Bovine skin splits	Composite film/Collagen applications	Improvement of mechanical, thermal properties and water solubility.	[70]
HAS	25	3D printing (SFFF)	Hydroxy apatite	Composite Scaffold Bone tissue	Enhanced mechanical properties	[79]
Amylomaize	80	Extrusion	Glycerol	Tissue engineering	Low tissue response of the host, due to degradation of amylomaize	[25]
Acetylated/hydroxypropylate HASs	-	Casting and blending	Ethylcellulose	Drug delivery	Potential sites specific for coating colon	[78]

## 7. Conclusions

Developmental efforts have been attempted over many decades to use starch as a raw material for a wide range of applications. However, mainly due to its inefficient suboptimal mechanical behavior, difficulty in processability, and hydrophilicity, still limit the application of starches for many industrial uses. From the studies described in this review, it is evident that there are possibilities for the utilization of HAS providing some unique functionalities in a wide array of diverse applications. Different HAS polymers can provide solutions for a wide range of food, bio-composite, and biomedical applications. Most importantly, HAS can typically readily drop in at existing processing systems as a bulk, be easily available raw, biopolymer that can be fabricated into form of powder, foams, hydrogels and fibers and as a compatible component in bio composites. The review provides a comprehensive report on the HAS properties and the latest updates in different fields, which will further open up a plethora of challenges that need to be addressed by the research community.

## Figures and Tables

**Figure 1 polymers-14-01235-f001:**
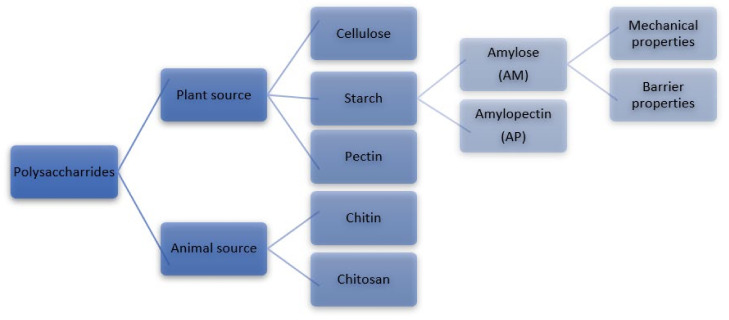
Classifications of polysaccharides.

**Figure 2 polymers-14-01235-f002:**
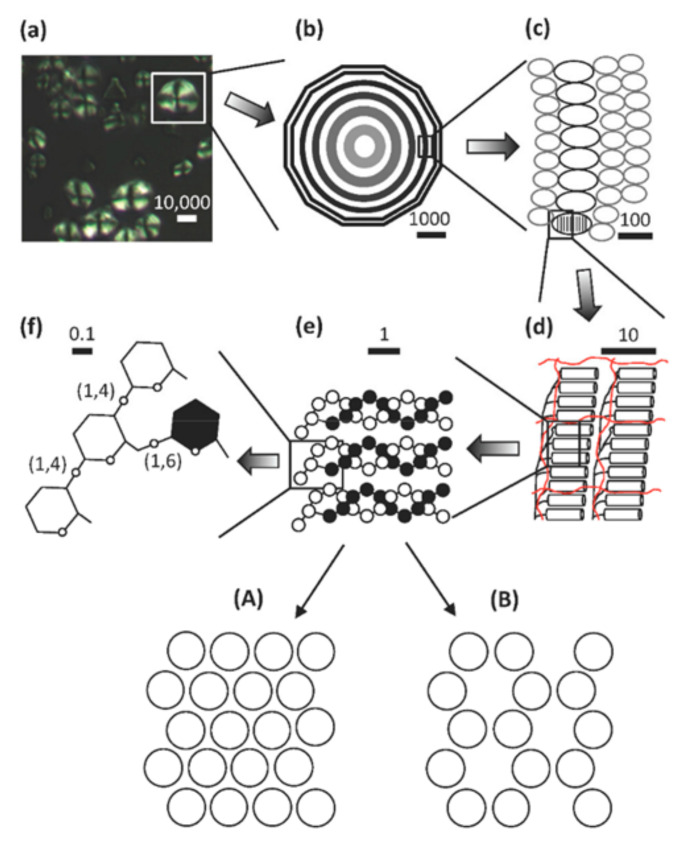
Dimensions in starch from granules to glycosyl units. (**a**) Maize starch granules observed under polarized light showing the “Maltese cross”, which indicates the radial organization within the starch granule. (**b**) A hypothetical granule with growth rings extending from the hilum. (**c**) Blocklets in semi-crystalline (black) and amorphous (grey) rings. (**d**) Crystalline and amorphous lamella formed by double helices (cylinders) and branched segments of AP (black lines). (**e**) Three double helices of AP, the double helices form either (**A**) or (**B**) polymorphic crystals. (**f**) Glucosyl units showing α-(1, 4) and α-(1, 6) linkages at the base of the double-helices (reproduced from [3]).

**Figure 3 polymers-14-01235-f003:**
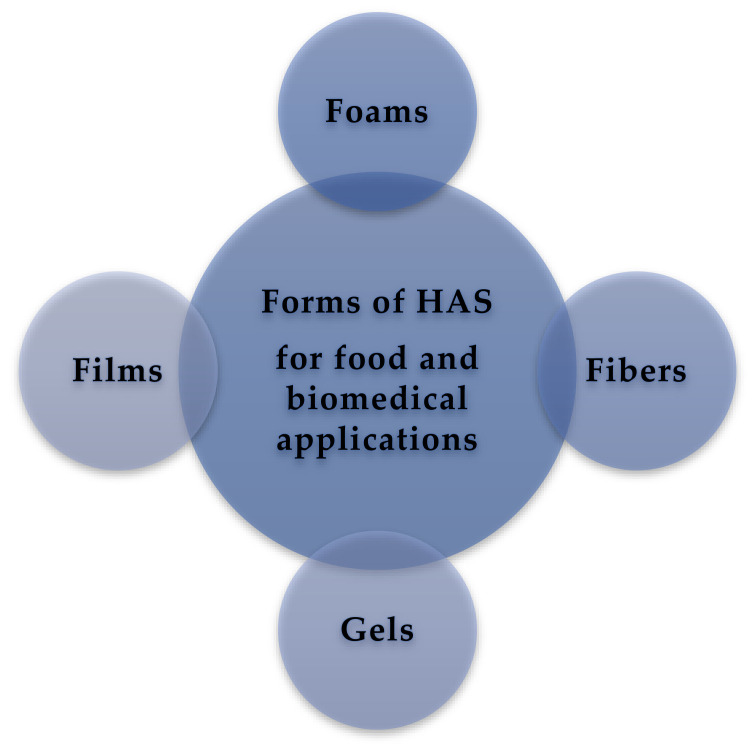
Forms and applications of HAS.

**Figure 4 polymers-14-01235-f004:**
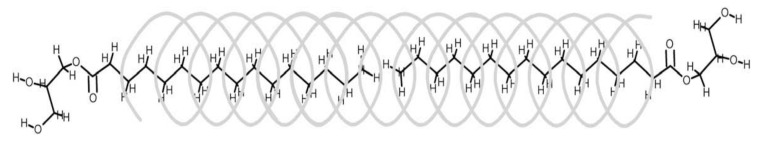
Schematic representation of a complex of AM with two monopalmitin molecules (reproduced with permission from Elsevier [50]) or Reprinted from Food hydrocolloids, Vol 23/1527–1534, Les Copeland, Jaroslav Blazek, Hayfa Salman, Mary Chiming Tang, Form and functionality of starch, Page No.5, Copyright (2022), with permission from Elsevier.

**Figure 5 polymers-14-01235-f005:**
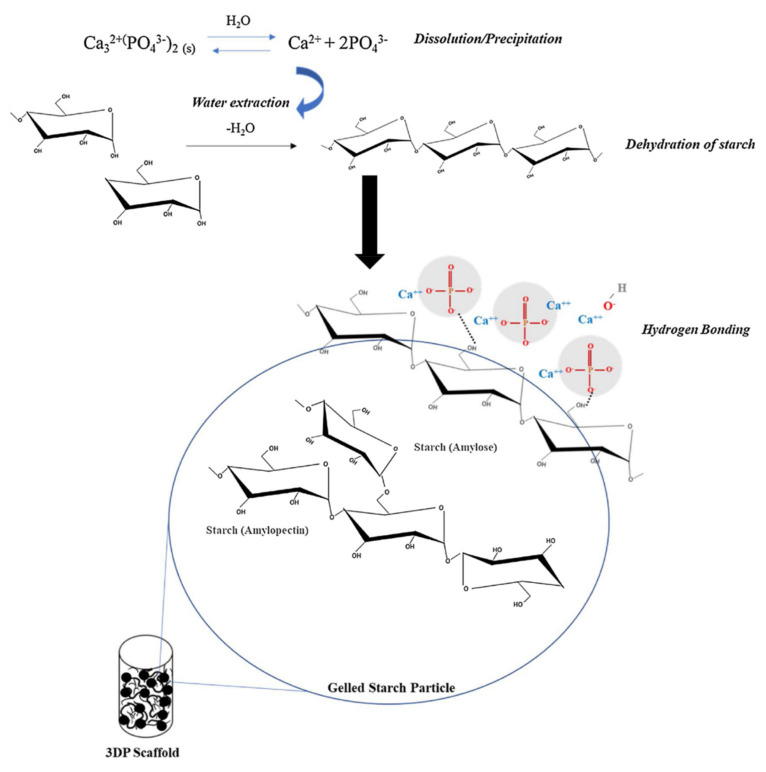
Reinforcing mechanism of starch based 3 DP scaffolds through particle and crystal interlocking: The dissolution and precipitation of calcium phosphate leads to setting and hardening of the starch (reproduced with permission from Elsevier [79]). Or: Reprinted from Additive Manufacturing, Vol 30/100817, Caitlin Koski; Susmita Bose, Effects of amylose content on the mechanical properties of starch hydroxyapatite 3D printed bone scaffolds, Page No.7, Copyright (2022), with permission from Elsevier.

**Table 1 polymers-14-01235-t001:** Mechanical properties of common plasticizers with HAS films.

Starch	AM%	Plasticizer Content %	Mechanical Properties TS (MPa) E (MPa)		EAB %	References
AM-only	99	15% glycerol	27 *	2200 *	2.8 *	[10]
Amylomaize	70	20% glycerol	ND	83	ND	[25]
HAS	>51	30% glycerol	2.04	11.83	0.24	[27]
HAS	>51	30% urea formamide	2.02	9.94	0.97	[27]
Gelose 80	82.9	9% [Emim] [OAc]	37 *	1180	12 *	[28]
Gelose 80	82.9	9% glycerol	36 *	1000	14 *	[28]
Corn Starch	80	20% glycerol	30.65	1079.67	4.60	[26]
Corn starch	80	20% Xylitol	37.10	1177.57	4.03	[26]
Corn starch	80	20% glycerol + Xylitol	37.29	1127.79	4.10	[26]

* obtained from figures in the corresponding references; [Emim] [OAc]: 1-ethyl-3-methylimidazolium acetate; ND: not determined.

## Data Availability

Exclude this statement.

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
