# Peer review of "High Amylose-Based Bio Composites: Structures, Functions and Applications"

_polymers, 2022, doi:10.3390/polym14061235_

Round 1

Reviewer 1 Report

This review provides a concise discussion on HAS applications and detailed forms relevant to its functionality with emphasis on biomaterials and its interactions with other natural polysaccharides. The authors introduced structure, constitute, physical and chemical properties of HAS and relevant production technology and protocols. Especially, it emphasizes the detailed information of HAS film, foam, fiber and gel, which would be the key points of their furture application. Moreover, more structural information should be added.

Author Response

Dear Reviewer,

Thank you for revising my manuscript and your comment about more structural information should be added.  

Information on the hierarchal structure of starch has been added to the introduction part, from line 66 supplemented with descriptive figure ''Figure 2'' about starch structure.

Thank you.

Reviewer 2 Report

The work is interesting and well written. Provides a lot of valuable information which are not widely known even among. There are only several shortcomings that I recommend to eliminate before publication.

First of all, there are a lot of acronyms. Even people familiar both with starch and polymers composites are not always able to decode them immediately. Therefore I suggest to supplement the manuscript with the list of abbreviations used.

Secondly, two sentences are confusing, i.e.:

Lines 56-57: “For example, an RNA interference line producing high yield of virtually pure A has been generated” – what does it mean “A”?

Lines 209-210: “Using exieting protocols, HAS is not completely gelatinized when bubbles are introduced to the gel and does not have strength to trap the bubbles and create the pores” – exieting seems to be a Chinese word.

And last but not least – the title of subsection 6: “Applications of HAS films”. The word “films” seems to be overused as in the subsections 6.1. and 6.2. the story is not about films.

Author Response

Dear Reviewer,

Thank you for revising my manuscript and for your comments.

Kindly find my response below:

1-First of all, there are a lot of acronyms. Even people familiar both with starch and polymers composites are not always able to decode them immediately. Therefore, I suggest to supplement the manuscript with the list of abbreviations used.

Response: A list of abbreviations has been added before the introduction part.

2-Secondly, two sentences are confusing, i.e.:

Lines 56-57: “For example, an RNA interference line producing high yield of virtually pure A has been generated” – what does it mean “A”?

Response: This has been corrected to AM at line 102.

Lines 209-210: “Using exieting protocols, HAS is not completely gelatinized when bubbles are introduced to the gel and does not have strength to trap the bubbles and create the pores” – exieting seems to be a Chinese word.

Response: This is has been corrected to exciting at line 254.

And last but not least – the title of subsection 6: “Applications of HAS films”. The word “films” seems to be overused as in the subsections 6.1. and 6.2. the story is not about films.

Response: The title has been changed to ''Uses of HAS''.

Thank you for your time and effort.